# Transcriptotype-Driven Discovery of Apigenin as a Therapy against Cholestatic Liver Fibrosis: Through Inhibition of PANoptosis and Following Type-I Interferon Responses

**DOI:** 10.3390/antiox13030256

**Published:** 2024-02-20

**Authors:** Shuni Duan, Xin Li, Junsong Han, Yang Yang, Ranyi Luo, Yajie Cai, Xiaojiaoyang Li, Qi Zheng, Jincheng Guo, Runping Liu

**Affiliations:** 1School of Chinese Materia Medica, Beijing University of Chinese Medicine, 11 Bei San Huan Dong Road, Beijing 100029, China; mayday111sy@163.com (S.D.); lixinbjzyydx@163.com (X.L.); qsl1531081647@163.com (Y.Y.); caiyajie134@163.com (Y.C.); bucmszzq@163.com (Q.Z.); 2Department of Nephrology, Dongzhimen Hospital, Beijing University of Chinese Medicine, Beijing 100007, China; hanjs21@163.com; 3School of Life Sciences, Beijing University of Chinese Medicine, Beijing 100029, China; ranyi_luo@163.com (R.L.); xiaojiaoyang.li@bucm.edu.cn (X.L.); 4School of Traditional Chinese Medicine, Beijing University of Chinese Medicine, Beijing 100029, China; ashin7597@163.com

**Keywords:** cholestatic liver fibrosis, transcriptotype-based drug screening, PANoptosis, type-I IFN, inflammation, apigenin

## Abstract

Cholangiopathies lack effective medicines and can progress into end-stage liver diseases. Mining natural product transcriptome databases for bioactive ingredients, which can reverse disease-associated transcriptomic phenotypes, holds promise as an effective approach for drug discovery. To identify disease-associated transcriptomic changes, we performed RNA-sequencing on bile duct ligation (BDL)-induced cholestatic liver fibrosis mice, as well as PBC and PSC patients, and found that PANoptosis and activation of type-I interferon (IFN) signaling were observed in BDL mice and patients with PBC and PSC. We then established a transcriptotype-driven screening system based on HERB and ITCM databases. Among 283 natural ingredients screened, apigenin (Api), which is widely distributed in varieties of food and medicinal plants, was screened out by our screen system since it reversed the expression pattern of key genes associated with PANoptosis and type-I IFN responses. In BDL, *Abcb4*^−/−^, and DDC-fed mice, Api effectively ameliorated liver injuries, inflammation, and fibrosis. It also protected cholangiocytes from bile acid-stimulated PANoptosis, thus alleviating damage-associated molecular pattern-mediated activation of TBK1-NF-κB in macrophages. Additionally, Api directly inhibited type-I IFN-induced downstream inflammatory responses. Our study demonstrated the pathogenic roles of PANoptosis and type-I IFN signaling in cholestatic liver fibrosis and verified the feasibility of transcriptotype-based drug screening. Furthermore, this study revealed a novel anti-inflammatory mechanism of Api and identified it as a promising candidate for the treatment of cholestatic liver fibrosis.

## 1. Introduction

Cholestatic liver diseases, including primary biliary cholangitis (PBC) and primary sclerosing cholangitis (PSC), are characterized by impairment of bile formation and/or flow that progress to hepatitis, liver fibrosis, and eventually cirrhosis and liver failure [1]. Cholangiocytes are the primary targets in cholangiopathies. Cholestasis disrupts the normal function of the bile duct epithelium and activates cholangiocytes, promoting ductular reaction and peritubular fibrosis. Cholangiocytes not only maintain biliary proliferation but also participate in complex crosstalk between various hepatic resident and recruited cells, playing a critical role in immune response and liver inflammation [2,3]. Due to its progressive nature and unclear pathogenesis, there is currently a lack of therapeutic drugs for cholestatic liver diseases, and liver transplantation remains the major curative option for end-stage patients. Therefore, novel therapy approaches to treat cholestatic liver diseases need to be developed.

Regulated cell death (RCD), mainly including pyroptosis, apoptosis, and necroptosis, is essential for the maintenance of tissue homeostasis in multicellular organisms [4]. Recent studies have revealed widespread crosstalk between different forms of RCDs [5,6,7], hence, the concept of PANoptosis was proposed by Malireddi et al. in 2019 [8]. PANoptosis is driven by the PANoptosome complex, which provides a multiprotein platform for the assembly of key components of pyroptosis, apoptosis, and necroptosis [9]. A recent study indicated that the hepatotoxicity of triptolide was mediated by the induction of PANoptosis in macrophages through increased mitochondrial dysfunction and reactive oxygen species (ROS) production [10]. Additionally, liproxstatin-1 (LPT1, ferroptosis inhibitor) was shown to alleviate steatosis and steatohepatitis in mice fed a high-fat, high-fructose diet, which may involve the inhibition of PANoptosis [11]. These findings collectively identified the critical roles of PANoptosis in hepatic homeostasis. Type-I interferon (IFNα and IFNβ) binding to the type-I IFN receptors (IFNARs) activates the JAK-STAT pathway, resulting in the activation of downstream signaling pathways and IFN-stimulated genes (ISGs). Type-I IFN signaling, which could be triggered by danger-associated molecular patterns (DAMP), occupies an important role in innate immune defense. Accumulated evidence has illustrated that type-I IFN signaling may promote hepatic disease progression [12,13]. Since PANoptosis is a relatively new concept, there are few reports regarding the role of PANoptosis in the regulation of type-I IFN. While apoptosis is shown to enhance type-I IFN production through activating cyclic guanosine monophosphate (GMP)-AMP synthase (cGAS)-stimulator of interferon genes (STING) pathway [14]. Additionally, previous studies have identified that ISGs product Z-DNA Binding Protein 1 (ZBP1), the critical mediator of PANoptosis, could bind dsDNA and recruit and phosphorylate TANK-binding kinase 1 (TBK1) and interferon regulatory factor 3 (IRF3) to initiate the transcription of IFN-I gene [15,16]. Overall, these results provide a strong basis for suppressing PANoptosis-triggered type-I IFN signaling as a promising treatment for liver diseases. However, the molecular mechanisms underlying PANoptosis and type-I IFN in cholestatic liver diseases remain poorly understood and require further investigation.

Natural products provide advantages for the treatment of hepatic disorders for their multi-target, multi-channel, and safety characteristics. Previous studies have highlighted the hepatoprotective activity of natural products profiled with anti-pyroptosis, apoptosis, or necroptosis properties, as well as ingredients with type-I IFN-inhibiting activities, such as baicalin, berberine, and curcumin [17,18]. In recent years, there has been an accumulation of natural product transcriptomics databases published and available online, providing novel strategies for the discovery of new natural product-derived drug candidates against cholestatic liver fibrosis.

In the current study, we aim to elucidate the role of PANoptosis and type-I IFN signaling on cholestatic liver fibrosis based on the RNA sequencing analysis, screening bioactive ingredients downregulating key genes of PANoptosis and type-I IFN from public natural product transcriptomics databases, and further investigate the therapeutic effects of apigenin (Api), the filtered-out candidate, on cholestatic liver fibrosis and underlying mechanisms.

## 2. Materials and Methods

### 2.1. Materials

Apigenin and SR-717 were obtained from Source Leaf Biological Technology (Shanghai, China). Recombinant human IFNα2a was purchased from Novoprotein Scientific (Shanghai, China). Poly (I:C) was purchased from MCE (Monmouth Junction, NJ, USA). Lipopolysaccharide (LPS) was purchased from Biorigin Biotechnology (Beijing, China). Assay kits for TUNEL and ROS were obtained from Beyotime Biotechnology (Nanjing, China). YO-PRO-1 (YP-1) dye and PI dye were obtained from Beyotime Biotechnology (Nanjing, China). Test kits for alanine aminotransferase (ALT), aspartate aminotransferase (AST), alkaline phosphatase (ALP), gamma-glutamyltransferase (GGT), lactate dehydrogenase (LDH), total bile acid (TBA), total bilirubin (TBIL), malondialdehyde (MDA), superoxide dismutase (SOD), and glutathione (GSH) were purchased from Nanjing Jiancheng Bioengineering Institute (Nanjing, China). Sirius red staining Kit was obtained from Solarbio Biotechnology (Beijing, China). QIAamp DNA Blood Mini kit was obtained from QIAGEN company (Hilden, Germany). Primary antibodies targeting Cytokeratin 19 (CK19, 10712-1-AP), caspase 3 (Casp 3, 19677-1-AP), mixed lineage kinase domain-like protein (MLKL, 66675-1-lg), interleukin-1β (IL-1β, 16806-1-AP), translocase of outer membrane 20 (TOM20, 11802-1-lg), janus kinase 1 (JAK1, 66466-1-lg), tyrosine kinase 2 (TYK2, 67411-1-lg), nuclear factor kappa B (NF-κB p65, 10745-1-AP), apoptosis-associated speck-like protein containing a CARD (ASC, 10500-1-AP), and β-actin (66009-1-lg) were obtained from Proteintech (Rosemont, IL, USA). The antibody against p-MLKL (AB187091) was purchased from Abcam (Berlin, Germany). Anti-DNA antibody (CBL186) was obtained from Millipore (Darmstadt, Germany). First antibodies included p-JAK1 (AP0530), p-TYK2 (AP0543), signal transducer and activator of transcription 1 (STAT1, A12075), p-STAT1 (AP0453), STAT2 (A3588), and p-STAT2 (AP0284) were purchased from ABclonal (Beijing, China). Antibodies detecting phosphorylated receptor-interacting protein kinase 3 (p-RIPK3, 91702), TBK1 (38066S), and p-TBK1 (5483S) were purchased from Cell Signaling Technology (Danvers, MA, USA). Antibodies against ZBP1 (sc-271483), Casp 8 (sc-81656), Casp 1 (sc-392736), RIPK3 (sc-374639), and gasdermin D (GSDMD, sc-393656) were purchased from Santa Cruz (Dallas, TX, USA). The secondary antibodies used were HRP-goat anti-rabbit IgG (abs20040) from ABSIN Bioscience (Shanghai, China), HRP-goat anti-mouse IgG (SA00001-1) from Proteintech (Rosemont, IL, USA), Alexa Fluor 594 anti-rabbit IgG (8889S) from Cell Signaling Technology (Danvers, MA, USA), and Alexa Fluor 488 anti-mouse IgG (A32723) from Thermo Fisher Scientific (Waltham, MA, USA).

### 2.2. Animal Study

C57BL/6J mice (8 weeks, 20–22 g, male) were obtained from SPF Biotechnology (Beijing, China). *Abcb4^−/−^* mice with C57BL/6N background were purchased from Jackson Laboratories (Sacramento, CA, USA) and bred in homozygote condition. *Abcb4^−/−^* mice (5 weeks, male) were used in this study. All mice were raised in a temperature-controlled specific pathogen-free (SPF) environment with a 12 h light/dark cycle and had access to water and standard chow ad libitum. All animal procedures were approved by the Institutional Animal Care and Use Committee of the Beijing University of Chinese Medicine.

To establish and evaluate the success of the model of cholestatic liver fibrosis, the C57BL/6J mice were randomly assigned to 2 groups after 1-week adaptive feeding: (1) Sham group; (2) BDL group (n = 8 per group), where mice received either BDL or sham surgery for 1 week. For BDL surgery, mice were briefly anesthetized using isoflurane and operated on a warming pad with aseptic conditions. The common bile duct was separated and doubly ligated, followed by suturing of the peritoneum and abdominal skin with 5-0 silk sutures. The mice in the Sham group also underwent exposure of the common bile duct without ligation. Animals were sacrificed under isoflurane anesthesia to collect tissues 7 days after surgery.

To investigate the effect of Api on cholestatic liver fibrosis, BDL, *Abcb4^−/−^*, and 3.5-diethoxycarbonyl-1.4-dihydrocollidine (DDC)-induced mouse models were used. Api was suspended using 0.5% CMC-Na. In the BDL model, C57BL/6J mice were randomly divided into 4 groups after 1-week adaptive feeding: (1) Sham group; (2) BDL group; (3) BDL + Api (25 mg/kg) low dose group; (4) BDL + Api (50 mg/kg) high dose group (n = 8 per group). The doses of Api were determined based on previous studies [19,20]. All animals received either BDL or sham surgery for 1 week; among them, mice in groups (3) and (4) were orally administrated with Api for 3 days pre-surgery and another 4 days after 3 days post-surgery. Mice were sacrificed at 7 days after BDL. In *Abcb4^−/−^* model, 5-weeks male *Abcb4^−/−^* and age-matched wild type (Wt) C57BL/6N mice were used for experiments. Mice were divided into 4 groups: (1) Wt group; (2) *Abcb4^−/−^* group; (3) *Abcb4^−/−^* + Api (25 mg/kg) low dose group; (4) *Abcb4^−/−^* + Api (50 mg/kg) high dose group (n = 8 per group). Mice in groups (3) and (4) were administrated with Api once daily for 4 weeks, whereas mice in groups (1) and (2) received 0.5% CMC-Na. Animals were sacrificed to collect tissues at 9 weeks old. In the DDC model, similarly, mice were randomly divided into 4 groups after 1-week adaptive feeding: (1) control (Ct) group; (2) DDC group; (3) DDC + Api (25 mg/kg) low dose group; (4) DDC + Api (50 mg/kg) high dose group (n = 8 per group). Mice in groups (2)–(4) were fed a 0.1% DDC-supplemented diet for 3 weeks to induce cholestatic liver fibrosis. Mice in groups (3)–(4) were administrated with Api once daily. In comparison, mice in group (1) were fed the normal chow and received 0.5% CMC-Na. At 3 weeks after administration, animals were anesthetized with isoflurane and sacrificed.

### 2.3. Virtual Screening from the Public Natural Product Transcriptome Databases

Disease-associated differentially expressed genes (DEGs) were obtained from the leading edge subset from the Gene Set Enrichment Analysis (GSEA) plots of PANoptosis and type-I IFN gene sets in BDL vs. Sham. Ingredient-associated DEGs were obtained from ingredients with high-throughput transcriptome data in the HERB database (BenCaoZuJian, http://herb.ac.cn/, accessed on 5 May 2023) and a small molecular expression platform in Integrated Traditional Chinese Medicine (ITCM, http://itcm.biotcm.net/index.html, accessed on 15 May 2023). By intersecting the disease-associated DEGs with ingredient-associated DEGs, ingredients were ranked by matching score (if the ingredient down-regulate disease related DEGs, add one point; if up-regulate, deduct one point), and the target ingredient was identified with the highest score. An “ingredient-target” network was constructed with the top five active ingredients and their target genes using Cytoscape software (3.9.1).

### 2.4. RNA-Seq and Bioinformatic Analysis

Total liver RNA from sham and BDL mice was extracted with TRIzol reagent (Vazyme, Nanjing, China) and quantified with NanoDrop 2000 (Thermo Scientific, Waltham, MA, USA). Library construction and RNA-Seq was conducted by Novogene (Beijing, China) as described previously [21]. A volcano plot and heatmap were conducted to visualize DEG outcomes using R (4.3.0). Pathway enrichment analysis was performed using GSEA software (4.3.2). We also analyzed gene expression of PBC, PSC, and healthy control samples in the available Gene Expression Omnibus (GEO) database (https://www.ncbi.nlm.nih.gov/geo/, accession number: GSE 119600, accessed on 19 May 2023). Additionally, the RNA-Seq data of MDA-MB-231 cells (GSE 120550) and MDA-MB-468 cells (GSE 133968) available from GEO were used in this study.

### 2.5. Cell Culture and In Vitro Study

Human intrahepatic biliary epithelial (HIBEC) cells and myeloid leukemia mononuclear cells (THP-1) obtained from the Chinese Academy of Sciences were cultivated in MEM and RPMI-1640 with 10% fetal bovine serum (FBS) and 1% penicillin/streptomycin, respectively. For in vitro studies, HIBEC cells were placed into a 6-well plate. After pretreatment with Api (2.5, 5, 10 μM) for 1 h, HIBEC cells were stimulated with CDCA (200 μM) for 24 h, and cell lysate and supernatant were collected for subsequent study. THP-1 cells were firstly treated with PMA (50 ng/mL) for 48 h to obtain THP-1-derived macrophages. For conditioned medium experiments, cell-free conditioned medium from HIBEC cells was diluted with fresh RPMI-1640 culture medium containing 1% FBS in a ratio of 1:1.

### 2.6. Flow Cytometry Analysis

Cellular ROS generation was measured by flow cytometry. Briefly, HIBEC cells were placed into a 6-well plate and treated as indicated. Then, cells were harvested using trypsinization and exposed to 10 μM DCFH-DA (S0033S, Beyotime) for 20 min at 37 °C, followed by flow cytometry (FACS Calibur, BD Biosciences, San Jose, CA, USA) analysis.

### 2.7. Isolation and Quantification of DNA

Total DNA in the cell culture of HIBEC cells was extracted by Qiagen DNA minikit (51104, QIAGEN, Hilden, Germany) according to the manufacturer’s instructions and quantified using NanoDrop 2000 (Waltham, MA, USA).

### 2.8. Assessment of Cell Death

The stock solutions of propidium iodide (PI, C1062, Beyotime) and YP-1 (C2022, Beyotime) were diluted in a ratio of 1:20 and 1:1000 in PBS to final concentrations of 50 μg/mL and 1 μM, respectively. HIBEC cells were treated as indicated and then stained with a working solution for 20 min at 37 °C. The images were captured using a confocal laser scanning microscope (Olympus FV3000, Tokyo, Japan).

### 2.9. Quantitative Real-Time PCR (qRT-PCR)

Total RNA from liver and cells was extracted using TRIzol reagent (R401-01, Vazyme) and converted to cDNA (R323-01, Vazyme), qRT-PCR assays were performed with SYBR Green (Q511-02, Vazyme) and relative mRNA levels were normalized by *Hprt1*. Primer sequences are described in Appendix A
Appendix A.

### 2.10. Statistical Analysis

All data were expressed as mean ± SEM using GraphPad Prism (8.0.1). Statistical analysis was made using Student’s *t*-test in comparisons of two groups and one-way ANOVA with Tukey’s post hoc among multiple groups. *p* < 0.05 was considered as statistical significance.

## 3. Results

### 3.1. PANoptosis and Following Type-I IFN Signaling Promote Liver Injury in BDL Mice and Patients with PBC and PSC

Experimental cholestatic liver fibrosis was induced by BDL. Mice were sacrificed 7 days after sham or BDL surgery. As shown in Figure 1A, histological analysis revealed that BDL mice exhibited marked changes, including bile duct proliferation, inflammatory infiltration, and reconstruction of the hepatic lobule structure. Masson staining and immunohistochemistry (IHC) staining of CK19, a cholangiocyte marker, indicated a remarkable increase in periportal fibrosis and cholangiocyte proliferation after BDL (Figure 1A–C). Elevations in serum ALT, AST, and LDH activity, as well as TBA and TBIL levels, were observed and provided further evidence for liver fibrosis in BDL mice (Figure 1D,E). In addition, the mRNA expression of *Ck19*, fibrogenic genes (*Col1a1*, *Tgf-β*, *Acta2*, *Fn*), and inflammatory genes (*Il-1β*) was increased in the liver of BDL mice compared to sham mice (Figure 1F). The above results demonstrate that liver injury was successfully developed by BDL.

Next, we investigated the change in gene expression profiles in the liver of BDL and sham-operated mice with RNA-seq-based transcriptomic analysis. Principal component analysis (PCA) was used to visualize the distribution of transcriptomic data. As shown in Figure 2A, the PCA score plots demonstrated significant separation of the BDL group away from the Sham group, indicating a phenotypic change in BDL mice. A volcano plot was performed to show DEGs and indicated that a total of 4523 DEGs (adjust *p* value < 0.05) were identified after BDL (Figure 2B), among which 2665 were up-regulated (labeled in red), and 1858 were down-regulated (labeled in blue). Considering that immune response and PANoptosis play a crucial role in the progression of cholestatic liver fibrosis, we also performed GSEA analysis. The GSEA results indicated significant enrichment in two immune response-related pathways in BDL mice (Figure 2C). Additionally, PANoptosis (Figure 2D)- and type-I IFN (Figure 2F)-associated pathways and DEGs were enriched in the BDL group.

Furthermore, we also analyzed the transcriptomic data from a patient cohort of PBC and PSC from the GEO database (GSE 119600). As shown in Appendix A, the PCA separates PBC and PSC patients from a healthy control. The volcano plot displayed 363 up-regulated and 550 down-regulated genes of PBC patients and 628 up-regulated and 320 down-regulated genes of PSC patients away from the control group (Appendix A). The GSEA plots demonstrated that immune response-related pathways were significantly up-regulated in PBC and PSC patients (Appendix A). GESA plots of type-I IFN signaling in PBC and PSC patients revealed similar outcomes to those with BDL mice (Appendix A). Additionally, PANoptosis-related pathways and DEGs were enriched in PBC patients but not in patients with PSC, suggesting distinct pathogenesis of these two cholangiopathies (Appendix A). We subsequently produced a heatmap to analyze the differential expression of key genes related to PANoptosis and type-I IFN, and the results are shown in Figure 2E,G,H. We found that key pathway genes were up-regulated in BDL mice as well as PBC and PSC patients, suggesting that targeting PANoptosis and type-I IFN represent promising strategies for the treatment of cholestatic liver fibrosis.

### 3.2. Apigenin Is Screened out through Transcriptotype-Based Scoring System

Considering the high expression of PANoptosis and type-I IFN in BDL mice and patients with cholestatic liver diseases, we hypothesized that an active compound that inhibits PANoptosis and type-I IFN may serve as a potential drug candidate. Hence, we established a transcriptotype-based scoring system to screen bioactive natural ingredients. The screening strategy is illustrated in Figure 3. Specifically, ingredient-associated DEGs were obtained through two public natural product transcriptome databases: HERB and ITCM (Figure 3A). In addition, disease (cholestatic liver fibrosis in this study)-associated key DEGs were identified among leading-edge subsets contributing most to the enrichment for PANoptosis and type-I IFN pathway in the GSEA analysis (Figure 3B). Bioactive compounds were mined by matching DEGs associated with ingredient and disease and further ranked based on matching score (Appendix A). The “ingredient-target” network with the five top active ingredients and their targets was displayed in Appendix A. Api with potent inhibitory effects on key genes associated with PANoptosis and type-I IFN pathways was screened out from 283 compounds (Figure 3C). The heatmap plots indicated that Api treatment decreased the expression of key genes involved in these two pathways in MDA-MB-231 cells (GSE 120550) and MDA-MB-468 cells (GSE 133968) (Figure 3D).

### 3.3. Apigenin Ameliorates Cholestatic Liver Fibrosis in Different Mouse Models

To evaluate the effect of Api on cholestatic liver fibrosis, BDL-induced mice model was used (Figure 4A). The biochemical markers of cholestatic liver injury were evaluated, and the results showed that compared with the Sham group, mice in the BDL group showed a significant increase in the serum levels of ALT, AST, ALP, GGT, TBA, and TBIL, which were significantly reduced after Api treatment (Figure 4B–G). Oxidative stress is reported as an important event upstream of PANoptosis [22]. The results of oxidative stress-related indicators showed that the hepatic levels of MDA (product of lipid peroxidation) significantly increased while that of antioxidant SOD and GSH apparently decreased after BDL, and Api treatment reversed the above changes (Figure 4H–J). H&E and Sirius red staining revealed that Api attenuated liver injury and collagen deposition in BDL mice (Figure 4K,L). IHC staining of CK19 demonstrated that Api reduced BDL-induced bile duct proliferation (Figure 4K,M). Furthermore, treatment with Api significantly reduced the hepatic mRNA levels of *Ck19*, fibrogenic genes (*Col1a1*, *Tgf-β*, *Acta2*, *Fn*), and inflammatory gene (*Il-6*) in BDL mice (Figure 4N). Similar results were also found on *Abcb4^−/^^−^*- and DDC-induced cholestasis models (Appendix A). Together, these results suggested that Api exerted a protective effect on cholestatic liver fibrosis.

### 3.4. Apigenin Alleviates BDL-Induced Cholestatic Liver Fibrosis through Inhibition of PANoptosis and Type-I IFN Signaling

We subsequently examined the impact of Api on PANoptosis and type-I IFN signaling pathways. TUNEL staining was used to characterize apoptotic and pyroptotic cell in liver tissue sections. As shown in Figure 5A,B, the percentage of TUNEL-positive cells was markedly increased in the BDL group compared with the Sham group, which was reversed after the administration of Api. The results were further verified in cell death determined by LDH release (Figure 5C). We further explored the related molecular mechanisms. Western blot results demonstrated that, compared with the Sham group, the protein expression of ZBP1 (the key “switch” of PANoptosis), the initiator Casp 8, and downstream executioner Casp 3 (markers of apoptosis) were markedly increased in BDL mice. Additionally, phosphorylation of RIPK3 and MLKL was obviously elevated in the model group, indicating the occurrence of necroptosis. In addition to apoptosis and necroptosis, the activation of GSDMD, Casp 1, and IL-1β (markers of pyroptosis) was also significantly up-regulated in mice of the BDL group. However, the results were reversed by treatment with Api (Figure 5D–G). These data indicated that Api could inhibit PANoptosis induced by BDL. In addition, it is well known that the JAK-STAT pathway mediates signal transduction of type-I IFN receptor, we thus detected the expression of related proteins by Western blot. Figure 5H,I showed that Api significantly attenuated the phosphorylation of JAK1, TYK2, and STAT1/2, indicating the inhibition of type-I IFN pathway. Together, the above results suggested that Api protected against cholestatic liver fibrosis by inhibiting PANoptosis and the type-I IFN pathway.

### 3.5. Apigenin Alleviates CDCA-Induced Oxidative Stress and Subsequent PANoptosis in HIBEC Cells

Accumulating evidence suggests that cholangiocytes are the primary targets in the pathogenesis of cholangiopathies, damaged or proliferative cholangiocytes plays vital roles in immune responses via mediating the recruitment and activation of inflammatory cells [23,24]. Hence, we presented the hypothesis that cholangiocytes undergoing PANoptosis resulted in the activation of macrophages during BDL-induced cholangiopathies, and this process can be inhibited by Api. To support this hypothesis, we further conducted in vitro experiments. CDCA was proven to induce the production of intracellular ROS and the activation of NLRP3 inflammasome [25]. Therefore, we first assessed the impact of Api on cellular oxidative stress on CDCA-stimulated HIBEC cells. Parameters of oxidative stress, including MDA, SOD, and GSH, were detected, and the results showed that Api pretreatment apparently reversed the increase in MDA and decrease in GSH and SOD in CDCA-treated HIBEC cells (Appendix A). Additionally, we determined intracellular ROS production by flow cytometry using a ROS detection kit. As shown in Appendix A, we found that Api reduced the accumulation of intracellular ROS in CDCA-stimulated HIBEC cells. These results indicated that Api exerted protective effects against cellular damage by inhibiting oxidative stress.

To further confirm the effect of Api on PANoptosis, propidium iodide (PI)/YO-PRO-1 (YP-1) staining was performed, PI-positive indicated cellular necroptosis, pyroptosis, or necrosis, and YP-1-positive indicated cellular apoptosis or necroptosis [22]. Immunofluorescence staining showed that HIBEC cells underwent death in multiple ways (Figure 6A). Similar results were obtained by detecting LDH release in the culture medium (Figure 6B). We further detected the expression of PANoptosis markers. The results of Western blot showed that CDCA stimulation increased the expression of ZBP1, the cleavage of Casp 3 and 8 and the phosphorylation of RIPK3 and MLKL as well as the activation of GSDMD, Casp 1 and IL-1β in HIBEC cells, whereas Api suppressed the activation of these proteins, indicating that Api inhibited CDCA-induced PANoptosis in HIBEC cells (Figure 6C–F). In agreement, fluorescence staining further showed that the co-localization of components of PANoptosome complex, ASC with Casp 8 or RIPK3, occurred in the CDCA-treated HIBEC cells, indicating PANoptopsis, while pretreatment with Api inhibited ASC-speck formation and PANoptosome assembly (Figure 6G,H).

### 3.6. Conditioned Medium Derived from Damaged Cholangiocytes Activates Macrophage via Type-I IFN Pathway

For the evidence that activated cholangiocytes-provoked recruitment and activation of macrophages promotes cholestasis and injury during the progression of biliary diseases [26,27], we supposed that Api could suppress the activation of macrophages by inhibiting upstream PANoptopsis signaling of CDCA-treated HIBEC cells. Therefore, we introduced the supernatant of treated HIBEC cells as a conditioned medium (CM) for cultivating THP-1-differentiated macrophages (Figure 7A). Our results revealed that CM from CDCA-treated HIBEC cells (CM-CDCA) remarkably increased the gene expression of *Ifnb1* and ISGs (*Cxcl10* and *Ifit1*) and inflammatory genes (*Tnf-α*, *Il-6*, and *Ccl2*) in THP-1-differentiated macrophages, while CM-CDCA+Api significantly decreased the expression of relative genes (Figure 7B,C). Furthermore, we detected the expression of the type-I IFN-associated pathway-related proteins, indicating that CM-CDCA induced NF-κB nuclear translocation and phosphorylation levels of TBK1, JAK1, TYK2, STAT1/2, which were decreased in CM-CDCA+Api group (Figure 7D–F). In line with this, pretreatment with GSK8612 and p-XSC, potent inhibitors of TBK1 and NF-κB, suppressed the expression of *Ifnb1* and ISGs (*Cxcl10* and *Ifit1*) and inflammatory genes (*Tnf-α*, *Il-6*, and *Ccl2*) in CM-CDCA-treated THP-1-differentiated macrophages (Figure 7G,H). Taken together, these results suggested that damaged HIBEC induced by CDCA could promote type-I IFN activation and inflammatory reaction in macrophages via TBK1-NF-κB signaling, while this process can be inhibited by Api administration.

Recent research showed that DAMPs from apoptotic, necroptotic, and pyroptotic cells enhanced inflammatory responses [28,29,30]. To investigate whether DNA released by damaged cholangiocytes serves as a DAMP signaling molecule to mediate the activation of macrophages, total DNA in CM was extracted using QIAamp DNA Mini Kit, and the concentration of isolated DNA was detected using NanoDrop. As shown in Appendix A, Api administration reduced DNA release into the extracellular space in response to CDCA stimulation in HIBEC cells. In addition, the co-localization of TOM20 (mitochondrial marker) and anti-DNA by immunofluorescence staining suggested that CDCA promoted mitochondrial DNA (mtDNA) release from mitochondria in HIBEC cells, which were significantly suppressed by Api pretreatment, indicating that DNA released from damaged HIBEC cells undergoing oxidative stress and PANoptosis consist mainly of mtDNA (Appendix A).

### 3.7. Apigenin Directly Inhibits Type-I IFN-Mediated Inflammatory Responses in Macrophages

To further investigate the direct impact of Api on DAMPs-pattern recognition receptors (PRRs)-mediated type-I IFN signaling in macrophages. THP-1-differentiated macrophages were either treated with SR717 to trigger STING signaling, transfected with poly (I:C) to stimulate RIG-I activation, or challenged with LPS to activate TLR4 signaling. Api downregulated the gene expression of *Ifnb1* after poly (I:C) transfection (Figure 8C) while paradoxically up-regulated *Ifnb1* expression when stimulated with SR717 and LPS at a concentration of 2.5 μM (Figure 8A,E). On the other hand, Api pretreatment observably inhibited all stimuli-induced upregulation of *Cxcl10* and *Ifit1*, downstream genes of type-I IFN signaling (Figure 8A,C,E). We then evaluated the effect of Api on type-I IFN-mediated inflammatory responses, qRT-PCR results indicated that Api administration reduced the expression of inflammatory genes (*Tnf-α*, *Il-6*, and *Ccl2*) in THP-1-differentiated macrophages treated with SR717, poly (I:C) and LPS (Figure 8B,D,F). The above results suggested that the inhibiting effect of Api on type-I IFN responses in macrophages was stimuli-specific, and Api may suppress type-IFN downstream signaling mediated by IFNAR instead of upstream signaling of IFN production mediated by PRRs-IRF3 pathway. qRT-PCR results in Figure 8G demonstrated that recombinant human IFNα2a-induced ISG expression, which was IFNAR-dependent, was reduced by Api treatment, providing further support to this view. Overall, these findings indicated that Api can directly restrain type-I IFN-mediated inflammatory responses in macrophages.

## 4. Discussion

With the continuous development of bioinformatics and high-throughput sequencing technologies, natural product transcriptome databases, including HERB and ITCM, are emerging and used for identifying bioactive ingredients and predicting molecular targets of natural products. In the current study, we developed an efficient strategy for mining active ingredients through comparative transcriptome analysis, namely transcriptotype-driven drug screening. A total of 42 disease-associated DEGs (such as *Ripk1*, *Ripk3*, *Jak1*, and *Ifitm1*) were identified among leading-edge subsets of PANoptopsis and type-I IFN signaling via this screening strategy. By matching these diseases-associated DEGs with natural ingredient-associated DEGs, Api, with the highest score, was filtered out from 283 ingredients. Systematic evaluation of Api in mice with cholestatic liver fibrosis and CDCA-treated HIBEC cells further highlighted the therapeutic potential of Api and further validated the feasibility of our screening strategy. Physiological and pathological development are complex biological processes that are regulated by intricate protein–macromolecular interaction networks rather than a single molecule. Compared with traditional screening methods typically designed for a single target, our strategy actively screens for compounds that are capable for regulating multiple pathways and multiple targets. This may represent a more effective approach for drug development to avoid the limited efficacy of a single targeted drug due to pathway crosstalk and compensatory mechanisms, increasing the likelihood of drug candidates entering advanced development stages. Furthermore, by applying this virtual screening strategy without establishing specialized molecular or cell-based models, both the time and economic costs involved in drug discovery and development are significantly relieved. Unfortunately, this screening method relies on existing transcriptome databases, which currently contain transcriptomic data for only a small proportion of compounds, potentially limiting the discovery of new drugs. Even so, our screening strategy has broad application prospects due to the decreasing costs of omics research and the improvement of data accessibility.

Excessive accumulation of bile acids due to cholestasis could promote bile duct damage and inflammatory responses. A study from Hao et al. demonstrated that CDCA may act as a DAMP to induce intracellular ROS production and activate the NLRP3 inflammasome (indicating the occurrence of oxidative stress and pyroptosis) in macrophages, thereby promoting the development of the cholestasis-associated sepsis [25]. Studies have revealed that mitochondrial damage mediated by oxidative stress is an effective stimulus for mtDNA leakage, which contributes to the NLRP3 inflammasome activation and subsequent pore formation in the plasma membrane, leading to mtDNA release into the extracellular space [31]. The antioxidant activity of Api has been well-defined previously [19]. In this research, we demonstrated that Api alleviated oxidative stress and subsequent PANoptopsis in cholestatic liver and HIBEC cells, while one limitation of the present study is that how oxidative stress promotes PANoptopsis remains unclear and needs further research. In addition, reduced mtDNA release by Api treatment was further evidenced by measurement of DNA concentration in cell culture medium, as well as co-staining of DNA and TOM20, indicating the inhibition of damage signals.

It is worth noting that injured cholangiocytes play an active role in inflammatory response by secreting cytokines and chemokines, thus recruiting immune cells into the periductal microenvironment in the progression of all kinds of cholangiopathies, including mouse models mentioned above [32]. Moreover, damaged cholangiocytes could release endogenous DAMPs into the extracellular space to trigger sterile inflammation [27]. In the current study, the crosstalk between cholangiocytes and macrophages during the pathogenesis of cholangiopathies was achieved through the secretion of DAMPs (mainly mtDNA) from PANoptotic cholangiocytes. Interestingly, in turn, type-I IFN was shown to induce cellular pyroptosis, apoptosis, and necroptosis governed by ZBP1 [16,33], and it was found that IFN-β treatment potentiated PANoptosis in macrophages during viral infection, which was relied on ZBP1 and specifically its Zα2 domain [34], indicating that there is a feedback loop between PANoptosis and type-I IFN. Our current study only identified the effects of PANoptosis on the activation of type-I IFN signaling, and their exact interaction will be further elucidated in future research.

The physiological role of type-I IFN in inflammation remains unclear so far. Recent reports identified that by modulating the activation of various STAT complexes, type-I IFN regulated distinct gene-expression programs. STAT1 homodimers bind gamma-activated sequences (GASs) to induce the expression of pro-inflammatory genes (including *Irf3* and *Cxcl9*), whereas STAT3 induces that of anti-inflammatory genes, demonstrating a dual role of type-I IFN in regulating the inflammatory response [35,36]. Therefore, type-I IFN responses require strict control to maintain immune homeostasis. In addition, the pathogenic role of type-I IFN signaling in liver disease is controversial. It is reported that the sense of DNA induced by dead hepatocytes and related type-I IFN signaling in non-parenchymal cells (NPCs) deteriorated radiation-induced liver disease [12]. Additionally, type-I IFN caused oxidative stress in mice suffering from viral hepatitis, while the blockade of the IFN-I signaling ameliorated virus-induced liver injury [37]. The study from Araujo et al. demonstrated that in drug-induced liver injury, sterile cell death triggered massive DNA release. In response to accumulated DNA, liver NPCs aggravated oxidative stress through type-I IFN, thereby promoting the progress of liver injury [38]. Our study supported these findings, showing that the type-I IFN pathway was up-regulated in both BDL mice and patients with cholestasis. Specifically, following the recognition of cholangiocyte-derived DAMPs by PRRs, TBK1 activated not only IRF3 but also NF-κB pathways, which in turn initiated the transcription of IFN-I and inflammatory cytokines in macrophages. Consequently, type-I IFN response and NF-κB cascade pathway collaboratively enhanced the inflammatory milieu in macrophages. We also noted that type-I IFN signaling increased the expression of ISGs (*Cxcl10* and *Ifit1*) and chemokines (*Cxcl10* and *Ccl2*) to promote the recruitment of inflammatory cells and the shaping of the inflammatory environment. Furthermore, the hepatoprotective effect of Api was related to direct and indirect inhibition of type-I IFN signaling in macrophages, thereby exerting anti-inflammatory effects. Intriguingly, Api significantly, yet stimuli-dependently, inhibited the production of type-I IFNs while directly disturbing type-I IFN-induced downstream effects. Future research is required to clarify whether Api has a direct molecular target in the downstream signaling pathway after binding of IFN-I to IFNAR. However, some studies have drawn conflicting conclusions about the role of interferon in liver disorders. A study by Petrasek et al. indicated that type-I IFN exerted anti-inflammatory activities and protected against TLR9-associated mice hepatic injury [39]. In addition, type-I IFN produced from liver parenchymal cells in an IRF3-dependent manner was protective in alcoholic-induced liver injury by increasing IL-10 production and decreasing TNF-α production in NPCs [40]. Further investigations to uncover the interaction between type-I IFN and inflammation, as well as the in-depth mechanisms of type-I IFN signaling in liver disease, will shed novel light on the biological role of type-I IFN signaling and the discovery of new therapeutic strategies for hepatic diseases.

## 5. Conclusions

Taken together, we established a transcriptotype-based drug screening system. Api, a food-derived bioactive ingredient, was screened out from 283 natural ingredients for its ability to reverse the transcriptional pattern of PANoptosis and type-I IFN signaling-associated genes, which contributes to the progression of cholangiopathies. This study not only demonstrated the feasibility of our screen strategy on drug discovery for diseases lacking well-defined molecular targets but also highlighted a novel anti-inflammatory mechanism of Api and identified it as a promising candidate for the treatment of cholestatic liver fibrosis.

## Figures and Tables

**Figure 1 antioxidants-13-00256-f001:**
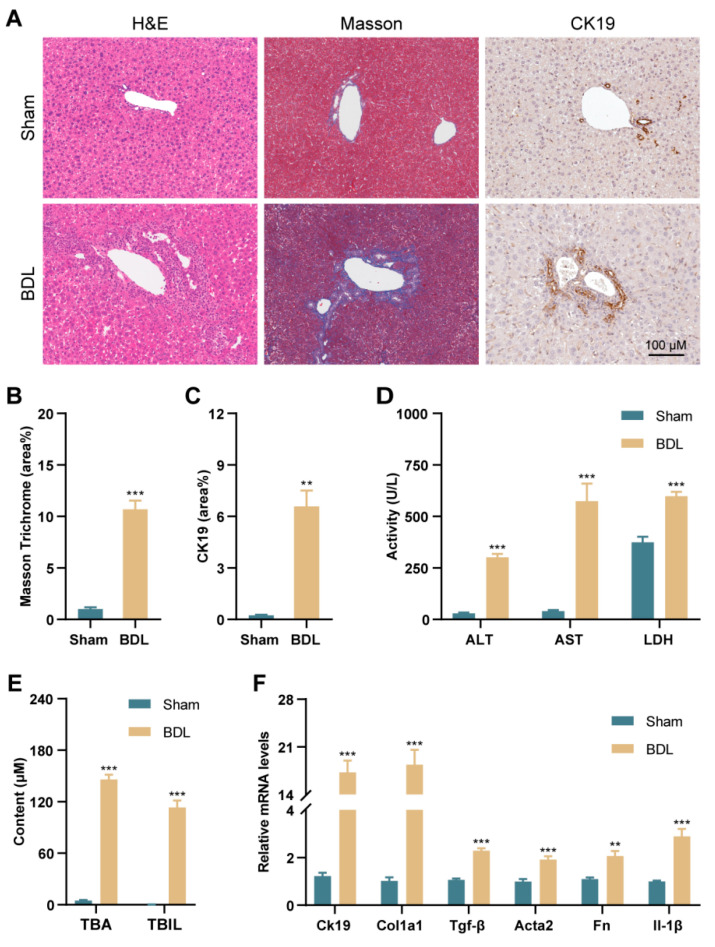
BDL-induced cholestasis mice model was successfully established. (**A**) H&E staining, Masson staining, IHC staining of CK19 in liver sections (scale bar = 100 µm). (**B**,**C**) Quantification of positive area for Masson and IHC staining. (**D**) Serum levels of ALT, AST and LDH. (**E**) Serum levels of TBA and TBIL. (**F**) The mRNA levels of *Ck19*, *Col1a1*, *Tgf-β*, *Acta2*, *Fn*, and *Il-1β* in liver tissues. *Hprt1* was used as an internal reference. ** *p* < 0.01, *** *p* < 0.001 vs. the Sham group (n = 6).

**Figure 2 antioxidants-13-00256-f002:**
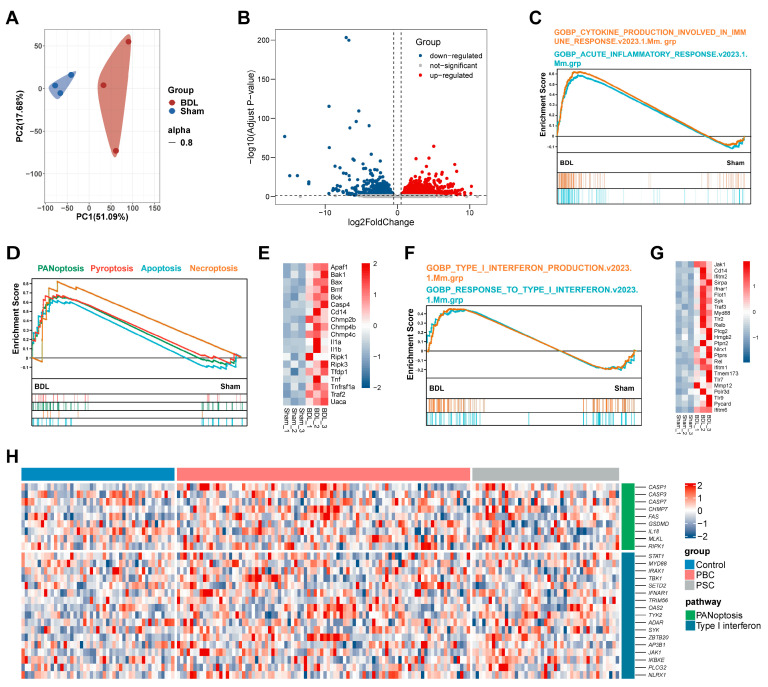
RNA-seq results analyses. (**A**) PCA plot of the RNA-seq results. (**B**) Volcano plot of DEGs in the liver of the BDL mice and the sham mice. (**C**) GSEA plot for the gene signature of cytokine production involved in immune response and acute inflammatory response pathway in livers of BDL mice compared with sham mice. (**D**) GSEA plot for the gene signature of PANoptosis, pyroptosis, apoptosis, and necroptosis pathway in livers of BDL mice compared with sham mice. (**E**) Heatmap of the key genes involved in PANoptopsis signaling as shown in (**D**). (**F**) GSEA plot for the gene signature of type-I IFN production and response to type-I IFN pathway in livers of BDL mice compared with sham mice. (**G**) Heatmap of the key genes involved in type-I IFN signaling as shown in (**F**). (**H**) Heatmap of the key genes involved in PANoptopsis and type-I IFN signaling in the blood of PBC and PSC patients compared with control group (GSE 119600).

**Figure 3 antioxidants-13-00256-f003:**
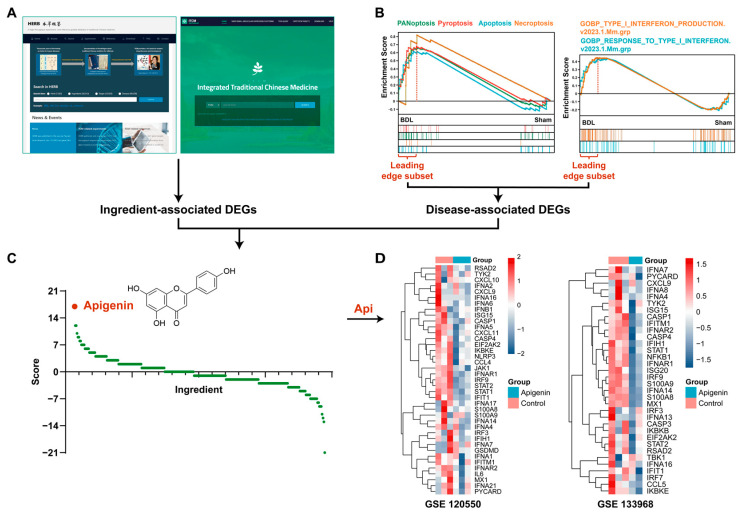
Flow chart of mining strategy in available natural product transcriptome databases. By searching natural products-derived transcriptome data in HERB and ITCM databases (**A**) with key genes in leading-edge subset of GSEA plot involved in PANoptosis and type-I IFN pathway (**B**), we gave +1 score if the *p* value < 0 (considered as down-regulation), otherwise we gave −1 score if the *p* value > 0 (considered as up-regulation), calculated and summed the scores to draw graphs (**C**). Api, as a potential ingredient with effective inhibitory activity, was ultimately screened. (**D**) Heatmap plots of key genes involved in PANoptopsis and type-I IFN signaling in MDA-MB-231 cells (GSE 120550) and MDA-MB-468 cells (GSE 133968) treated with Api or untreated.

**Figure 4 antioxidants-13-00256-f004:**
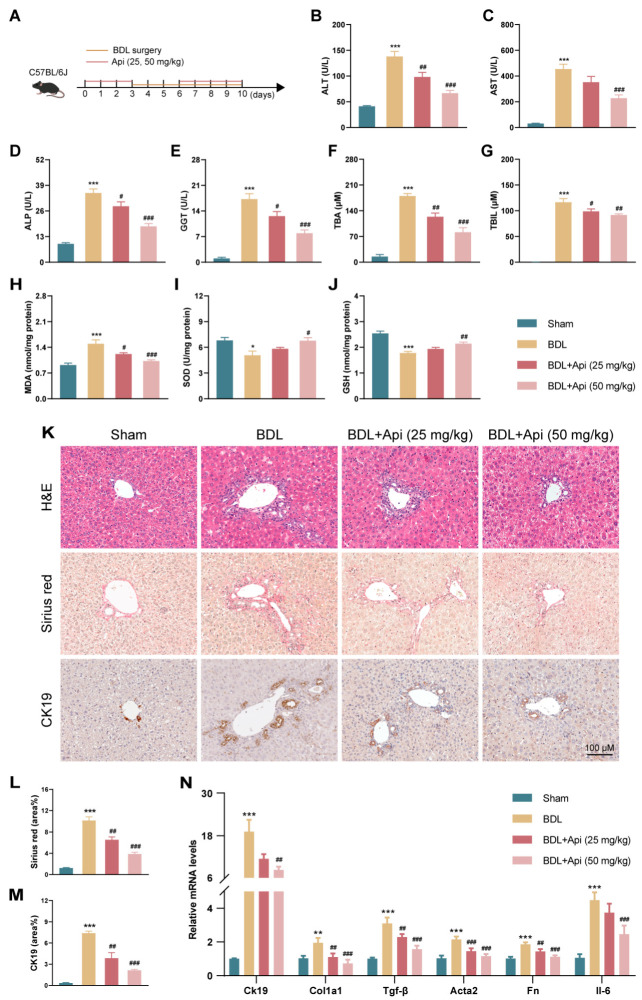
Effects of Api on BDL-induced hepatic fibrosis in mice. (**A**) The schematic diagram of mice experimental design. Serum levels of ALT (**B**), AST (**C**), ALP (**D**), GGT (**E**), TBA (**F**), and TBIL (**G**). Hepatic levels of MDA (**H**), SOD (**I**), GSH (**J**). (**K**) H&E staining, Sirius red staining, IHC staining of CK19 in liver sections (scale bar = 100 µm). Quantification of positive area for Sirius red (**L**) and IHC staining (**M**). (**N**) The mRNA levels of *Ck19*, *Col1a1*, *Tgf-β*, *Acta2*, *Fn*, and *Il-6* in liver tissues. *Hprt1* was used as an internal reference. * *p* < 0.05, ** *p* < 0.01, *** *p* < 0.001 vs. the Sham group, ^#^ *p* < 0.05, ^##^ *p* < 0.01, ^###^ *p* < 0.001 vs. the BDL group (n = 6).

**Figure 5 antioxidants-13-00256-f005:**
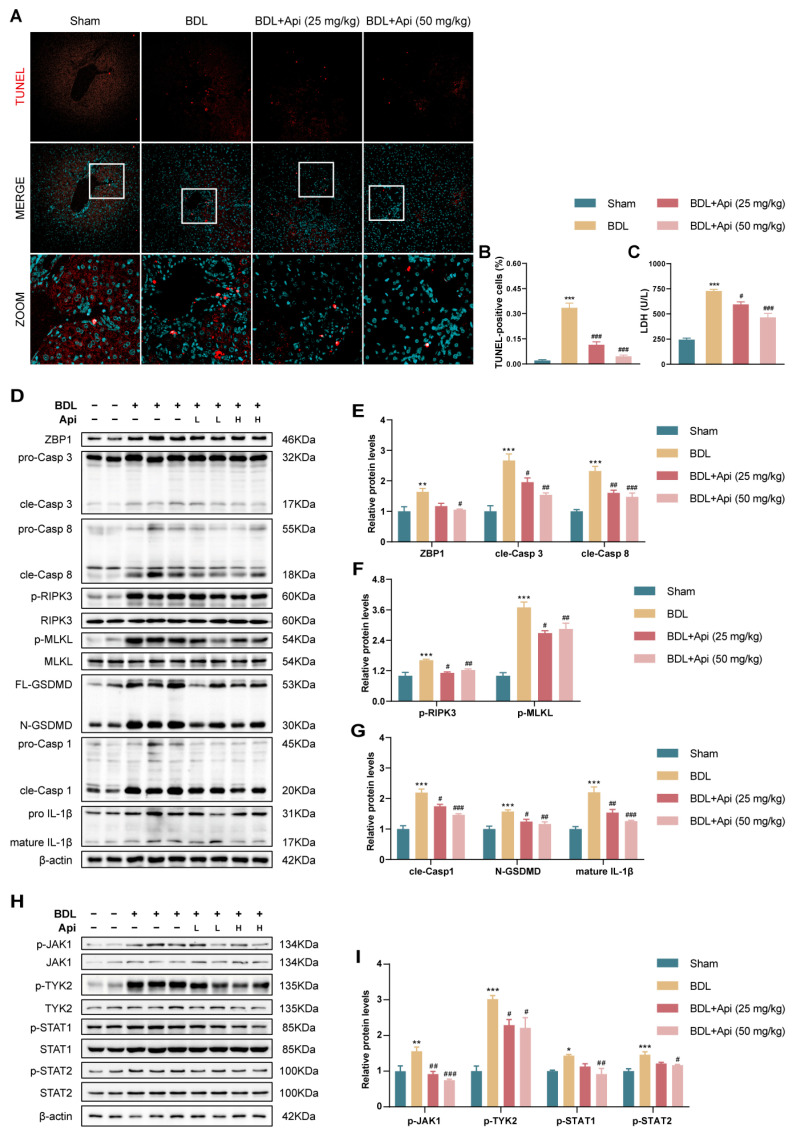
Effects of Api on PANoptosis and type-I IFN pathway in liver of BDL-induced fibrotic mice. (**A**) TUNEL staining of liver sections (magnification, ×40). Red: TUNEL-positive cells, blue: nucleus. Panel (zoom) represents enlarged area delineated by the white square box. (**B**) Percentage of TUNEL-positive cells. (**C**) Serum levels of LDH. (**D**–**G**) The hepatic protein levels of PANoptosis markers including ZBP1, Casp 3, cleaved Casp 3, Casp 8, cleaved Casp 8, phosphorylated RIPK3, total RIPK3, phosphorylated MLKL, total MLKL, GSDMD, cleaved GSDMD, Casp 1, cleaved Casp 1, Il-1β, and mature Il-1β determined by Western blot analysis. (**H**,**I**) The hepatic protein levels of type-I IFN markers, including phosphorylated JAK1, total JAK1, phosphorylated TYK2, total TYK2, phosphorylated STAT1, total STAT1, phosphorylated STAT2, and total STAT2, determined by Western blot analysis. * *p* < 0.05, ** *p* < 0.01, *** *p* < 0.001 vs. the Sham group, ^#^ *p* < 0.05, ^##^ *p* < 0.01, ^###^ *p* < 0.001 vs. the BDL group (n = 6).

**Figure 6 antioxidants-13-00256-f006:**
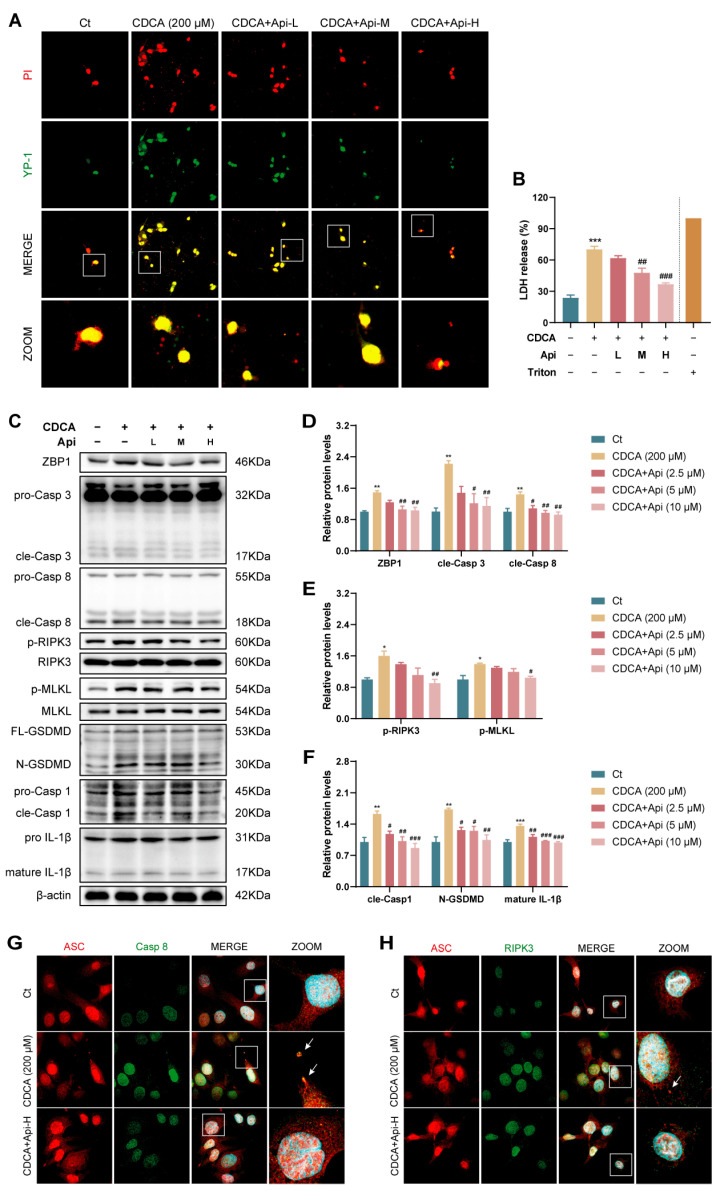
Effects of Api on PANoptosis in CDCA-stimulated HIBEC cells. (**A**) Fluorescence images of dual PI (red) and YP-1 (green) staining for HIBEC cells (magnification, ×100). Yellow represents co-localization, white square indicates area displayed in zoom panel. (**B**) Levels of LDH in culture medium. (**C**–**F**) The cellular protein levels of PANoptosis markers, including ZBP1, Casp 3, cleaved Casp 3, Casp 8, cleaved Casp 8, phosphorylated RIPK3, total RIPK3, phosphorylated MLKL, total MLKL, GSDMD, cleaved GSDMD, Casp 1, cleaved Casp 1, Il-1β, and mature Il-1β determined by Western blot analysis. (**G**) Fluorescence images of dual ASC (red) and Casp 8 (green) staining for HIBEC cells (magnification, ×100). Nuclei were stained using DAPI (blue), arrow points to a larger ASC speck, white square indicates area displayed in zoom panel. (**H**) Fluorescence images of dual ASC (red) and RIPK3 (green) staining for HIBEC cells (magnification, ×100). Nuclei were stained using DAPI (blue), arrow points to a larger ASC speck, white square indicates area displayed in zoom panel. * *p* < 0.05, ** *p* < 0.01, *** *p* < 0.001 vs. the Ct group, ^#^ *p* < 0.05, ^##^ *p* < 0.01, ^###^ *p* < 0.001 vs. the CDCA group (n = 3).

**Figure 7 antioxidants-13-00256-f007:**
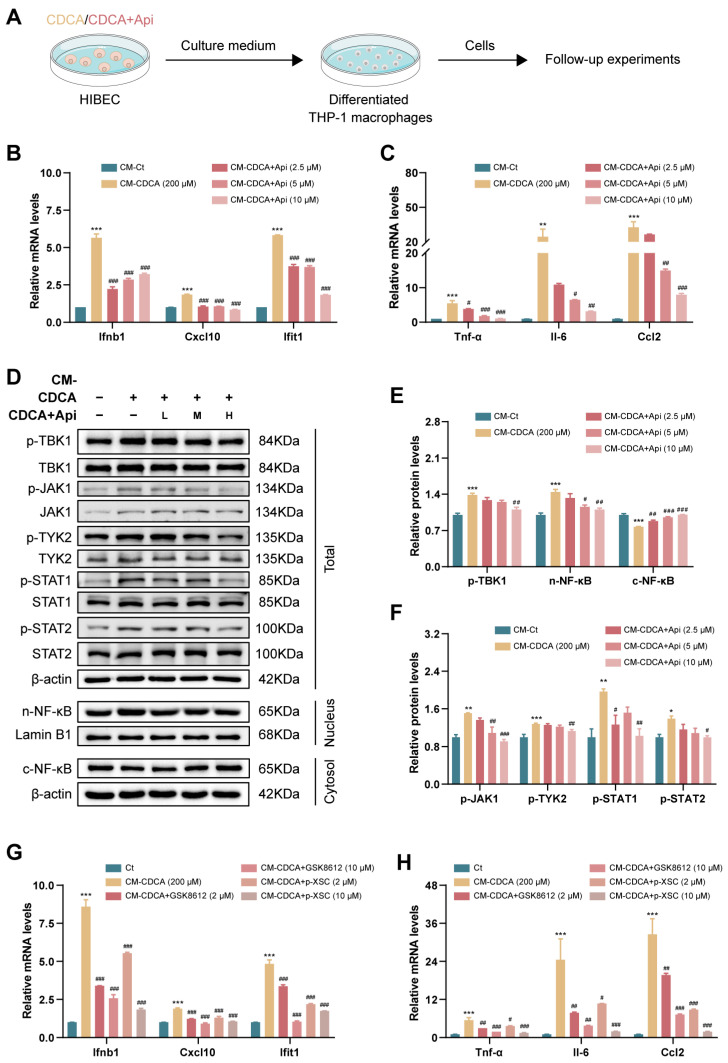
Effects of Api on type-I IFN pathway of THP-1-derived macrophages by alleviating PANoptosis in HIBEC cells. (**A**) Experimental protocol. The mRNA levels of *Ifnb1*, *Cxcl10*, *Ifit1* (**B**) and *Tnf-α*, *Il-6*, and *Ccl2* (**C**) in THP-1-derived macrophages treated with HIBEC conditioned medium. (**D**–**F**) The cellular protein levels of type-I IFN signaling markers, including phosphorylated TBK1, total TBK1, phosphorylated JAK1, total JAK1, phosphorylated TYK2, total TYK2, phosphorylated STAT1, total STAT1, phosphorylated STAT2, and total STAT2, nuclear and cytoplasmic NF-κB determined by Western blot analysis. The mRNA levels of *Ifnb1*, *Cxcl10*, *Ifit1* (**G**), *Tnf-α*, *Il-6*, and *Ccl2* (**H**) in THP-1-derived macrophages treated with CM from HIBEC and inhibitors of TBK1 (GSK8612) and NF-κB (p-XSC). *Hprt1* was used as an internal reference. * *p* < 0.05, ** *p* < 0.01, *** *p* < 0.001 vs. the CM-Ct group; ^#^ *p* < 0.05, ^##^ *p* < 0.01, ^###^ *p* < 0.001 vs. the CM-CDCA group (n = 3).

**Figure 8 antioxidants-13-00256-f008:**
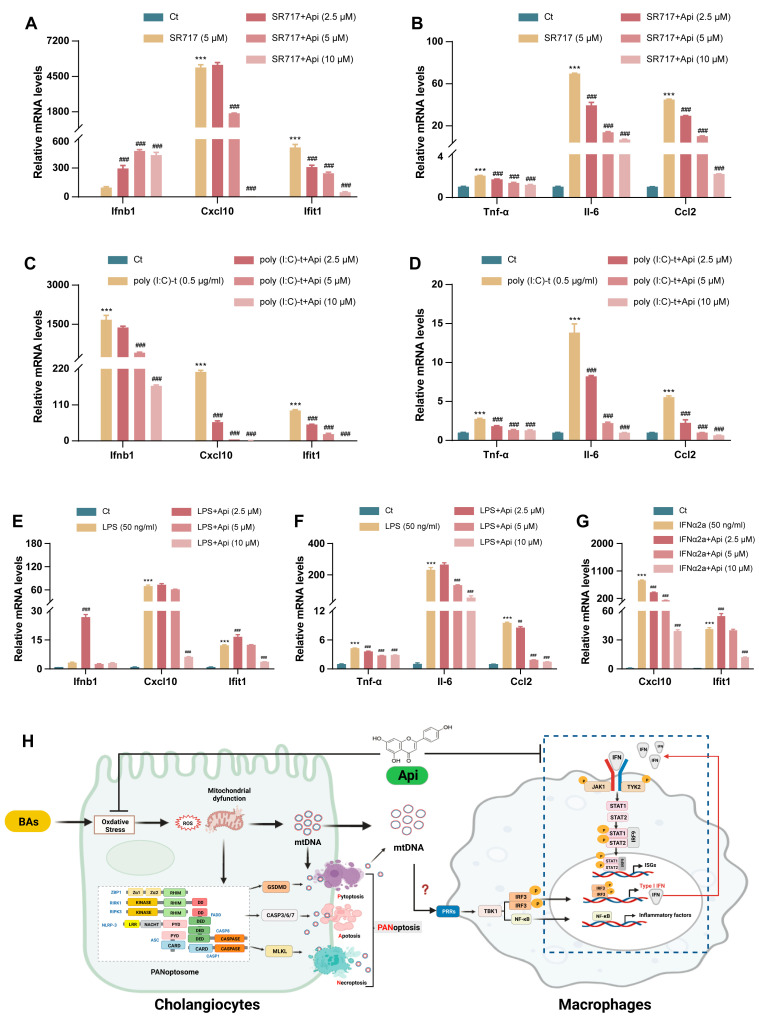
Effects of Api on type-I IFN signaling of THP-1-derived macrophages. The mRNA levels of *Ifnb1*, *Cxcl10*, *Ifit1* (**A**), *Tnf-α*, *Il-6*, and *Ccl2* (**B**) in THP-1-derived macrophages treated with SR717 (5 μM). The mRNA levels of *Ifnb1*, *Cxcl10*, *Ifit1* (**C**), *Tnf-α*, *Il-6*, and *Ccl2* (**D**) in THP-1-derived macrophages transfected with Poly (I:C) (0.5 μg/mL). The mRNA levels of *Ifnb1*, *Cxcl10*, *Ifit1* (**E**), *Tnf-α*, *Il-6*, and *Ccl2* (**F**) in THP-1-derived macrophages treated with LPS (50 ng/mL). The mRNA levels of *Cxcl10* and *Ifit1* (**G**) in THP-1-derived macrophages treated with IFNα2a (50 ng/mL). *Hprt1* was used as an internal reference. (**H**) Schematic diagram of hepatoprotective mechanism of Api against cholestatic liver fibrosis. RHIM: RIP homotypic interaction motif; DD: death domain; DED: death effector domain; LRR: leucine-rich repeat; PYD: pyrin domain; CARD: caspase recruitment domain. *** *p* < 0.001 vs. the Ct group; ^##^ *p* < 0.01, ^###^ *p* < 0.001 vs. the model group (n = 3).

## Data Availability

The data generated or analyzed in this study are available from the corresponding author upon reasonable request.

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
