# Peer review of "Transcriptotype-Driven Discovery of Apigenin as a Therapy against Cholestatic Liver Fibrosis: Through Inhibition of PANoptosis and Following Type-I Interferon Responses"

_antioxidants, 2024, doi:10.3390/antiox13030256_

Round 1
Reviewer 1 Report
Comments and Suggestions for Authors
The manuscript investigated the efficacy of apigenin, a flavonoid exhibiting therapeutic properties, as a modulator of hepatic cholestatic fibrosis due to its ability to inhibit PANoptosis and type 1 IFN signaling.
Apigenin was chosen through a Virtual screening of public natural product transcriptome databases. The study includes in vitro and in vivo experiments.
The research topic is interesting and the results could contribute to knowledge on the involvement of PANoptosis in the induction of hepatic cholestatic fibrosis with the aim of identifying new potential therapeutic targets and new natural drugs.
Critical points.
1. INTRODUCTION. This section needs to be shortened, particularly regarding the description of RCD and PANoptosis.
2) RESULTS. This section should include only the description of the reported results. The reasons for the experiments and the related bibliographical references must be avoided. Comments on results should be made only in the discussion to avoid repetition.
3) DISCUSSION. This section also needs to be shortened by eliminating repetitions (lines 498-490), overly general statements (lines 498-490; 509-516) and redundant bibliographic citations. Comments should focus on current results.
4) The authors should comment on the differences in the expression of PANoptosis-related genes in PBC patients and PSC patients. Also, statement on lines 268-269. seems to contradict the one in lines 276-277.
5) The limitations of the study musto to be added.
The manuscript requires very careful revision regarding the English language (there are several grammatical errors and incomplete sentences) and editing.
Comments on the Quality of English LanguageThe manuscript requires very careful revision regarding the English language (there are several grammatical errors and incomplete sentences) and editing.
Author Response
Reviewer’s comments
The manuscript investigated the efficacy of apigenin, a flavonoid exhibiting therapeutic properties, as a modulator of hepatic cholestatic fibrosis due to its ability to inhibit PANoptosis and type 1 IFN signaling.
Apigenin was chosen through a Virtual screening of public natural product transcriptome databases. The study includes in vitro and in vivo experiments.
The research topic is interesting and the results could contribute to knowledge on the involvement of PANoptosis in the induction of hepatic cholestatic fibrosis with the aim of identifying new potential therapeutic targets and new natural drugs.
Critical points.
- INTRODUCTION. This section needs to be shortened, particularly regarding the description of RCD and PANoptosis.
Response: Thank you for your valuable suggestion. We have shortened the section to reflect the most important messages. Thank you very much.
- 2. This section should include only the description of the reported results. The reasons for the experiments and the related bibliographical references must be avoided. Comments on results should be made only in the discussion to avoid repetition.
Response: Thank you for your valuable suggestion. We have shortened the section, and removed the repeated content. Thank you very much.
- 3. This section also needs to be shortened by eliminating repetitions (lines 498-490), overly general statements (lines 498-490; 509-516) and redundant bibliographic citations. Comments should focus on current results.
Response: Thanks for the reviewer’s kind suggestions. We have shortened the section, removed the repeated content in lines 498-490 and overly general statements in lines 498-490 and 509-516. Thank you very much.
- The authors should comment on the differences in the expression of PANoptosis-related genes in PBC patients and PSC patients. Also, statement on lines 268-269. seems to contradict the one in lines 276-277.
Response: Thanks for your comment. The differences in the expression of PANoptosis- related genes in PBC patients and PSC patients may attributed to distinct pathogenesis of these two cholestatic liver diseases. The relevant description was supplemented into the revised manuscript (line 273). In addition, results described in lines 268-269 and lines 276-277 are not contradictory, the former describes PCA results that represents sample clustering of RNA Seq results, indicating transcriptomic differences between PBC/PSC patients and the control group, while the latter is GSEA result showing the enrichment difference of PANoptosis pathways between PBC/PSC patients and control group.
- The limitations of the study must to be added.
Response: Thanks for the reviewer’s kind suggestions. The limitations of the study have described in the Discussion part (lines 501-505, 516-518).
- The manuscript requires very careful revision regarding the English language (there are several grammatical errors and incomplete sentences) and editing.
Response: Thanks for the reviewer’s kind suggestions. We have checked our manuscript very carefully again and English language usage and grammatical mistake has been edited.

Reviewer 2 Report
Comments and Suggestions for Authors
In this manuscript, the authors tried to find out the role of PANoptosis and type-I IFN signaling on cholestatic liver fibrosis based on RNA sequencing analysis. They screened 283 bioactive ingredients downregulating key genes of PANoptosis and type-I IFN from public natural product transcriptomics databases. Then they found the therapeutic effects of apigenin (Api) on cholestatic liver fibrosis. However, this reviewer has several concerns.
Major comments:
1. In Figure 3, how did the author select the Api from 283 compounds is unclear. If other candidates were analyzed, that information should be open to the public with Api.
2. In Figure 4, are there any changes in ALP, gGTP, and other biliary enzymes?
Regarding Figure 4J, a low magnification wide field of view needs to be analyzed.
3. The doses of Api for in vivo mice experiments were set at 25 mg/kg and 50 mg/kg. How were they determined?
4. Also, the doses of Api in vitro HIBEC cell experiments were set at 2.5, 5, and 10 mM. How were they determined?
Author Response
Reviewer’s comments
In this manuscript, the authors tried to find out the role of PANoptosis and type-I IFN signaling on cholestatic liver fibrosis based on RNA sequencing analysis. They screened 283 bioactive ingredients downregulating key genes of PANoptosis and type-I IFN from public natural product transcriptomics databases. Then they found the therapeutic effects of apigenin (Api) on cholestatic liver fibrosis. However, this reviewer has several concerns.
Major comments:
- In Figure 3, how did the author select the Api from 283 compounds is unclear. If other candidates were analyzed, that information should be open to the public with Api.
Response: Thank you for your valuable suggestion. A total of 283 compounds were ranked and plotted in Fig. 3 by scoring (we set +1 score if the target gene was down-regulated, otherwise gave -1 score if the target gene was up-regulated). A complete list was provided as Supplementary Table. S2. In the revised manuscript, we construct a “ingredient-target” network (Supplementary Fig. S2) using the top five active ingredients and their target genes by Cytoscape software to further improve clarity.
- In Figure 4, are there any changes in ALP, gGTP, and other biliary enzymes?
Response: Thank you for your valuable suggestion. We have supplemented the related experiments to detect the serum levels of ALP and GGT in Fig. 4D-E and in lines 309-310.
- Regarding Figure 4J, a low magnification wide field of view needs to be analyzed.
Response: Thank you for your valuable suggestion. The AperioVersa Scanner was used to scan the entire slide in this study as described in the Method section. However, magnified view near the portal area were used for better illustration in Fig. 4K. The staining quantification in Fig. 4L-M was based on the whole tissue. Response Figure 1 shows a low magnification wide field of view.
Response Figure 1. H&E staining, Sirius red staining, IHC staining of CK19 in liver sections.
- The doses of Api for in vivo mice experiments were set at 25 mg/kg and 50 mg/kg. How were they determined?
Response: Thank you for your comments and questions. We apologized that we did not clearly explain the reasons for choosing the drug dosages in the manuscript. Based on previous reports (30mg/kg in Reference (Zheng et al., 2021); 10/30/50mg/kg in Reference (Feng et al., 2016)), 25 and 50 mg/kg of Api were used in the current study. These details have been added in the revised paper (Lines 146-147).
Reference:
Feng, X., Weng, D., Zhou, F., Owen, Y. D., Qin, H., Zhao, J., . . . Shen, P. (2016). Activation of PPARγ by a Natural Flavonoid Modulator, Apigenin Ameliorates Obesity-Related Inflammation Via Regulation of Macrophage Polarization. EBioMedicine, 9, 61-76. doi: 10.1016/j.ebiom.2016.06.017
Zheng, S., Cao, P., Yin, Z., Wang, X., Chen, Y., Yu, M., . . . Yang, X. (2021). Apigenin protects mice against 3,5-diethoxycarbonyl-1,4-dihydrocollidine-induced cholestasis. Food Funct, 12(5), 2323-2334. doi: 10.1039/d0fo02910f
- 5. Also, the doses of Api in vitro HIBEC cell experiments were set at 2.5, 5, and 10 mM. How were they determined?
Response: Thank you for your comments and questions. In the early stage of this experiment, CCK8 experiment was conducted to determine the effect of Api on cell viability, and the results showed that Api had no significant effect on HIBEC cell viability at concentrations below 40μM. Therefore, the doses of 2.5, 5, and 10μM were selected in this study.

Round 2
Reviewer 1 Report
Comments and Suggestions for Authors
The authors answered to the comments and make required change in the manuscript that is now suitable for publication.
Comments on the Quality of English LanguageThe authors revised the manuscrip improving English language